# Temperature and radiative responses to anthropogenic aerosols over the Mediterranean Basin based on CMIP6 Earth system models

Alkiviadis Kalisoras<sup>1</sup>, Prodromos Zanis<sup>1</sup>, Aristeidis K. Georgoulias<sup>1,2</sup>, Dimitris Akritidis<sup>1</sup>, Robert J. Allen<sup>3</sup>, Vaishali Naik<sup>4</sup>

- Department of Meteorology and Climatology, School of Geology, Aristotle University of Thessaloniki, Thessaloniki, Greece Climate and Atmosphere Research Center, The Cyprus Institute, Nicosia, Cyprus
  - <sup>3</sup>Department of Earth and Planetary Sciences, University of California Riverside, Riverside, CA, USA
  - <sup>4</sup>NOAA Geophysical Fluid Dynamics Laboratory, Princeton, NJ, USA

Correspondence to: Alkiviadis Kalisoras (kalisort@geo.auth.gr)

Abstract. Here, we assess the amplification of near-surface warming in the Mediterranean (MED) resulting from global anthropogenic aerosol (AA) reductions, based on simulations from CMIP6 Earth system models (ESMs). Temperature and radiative responses are investigated over the MED. The effective radiative forcing (ERF) and near-surface temperature (TAS) exhibit decreasing trends until around 1980 followed by increasing trends, driven by air pollution control policies. The annual mean ERF at the top-of-atmosphere over the MED changes by 2.37±1.06 W m<sup>-2</sup> between the peak AA period (1970-1979) and the near-present period (2005-2014). During this interval, the annual mean TAS increases by 0.67±0.37 °C. Overall, the multimodel ensemble shows a robust amplification of warming over the MED on annual scale resulting from global AA reductions from 1970-1979 to 2005-2014, in good agreement with observational datasets over land. The model simulations indicate that AAs are responsible for 49% (39%) of the annual (summer) warming between the two periods. In the winter, ESMs produce an overestimated warming of 1.19 °C, with AAs contributing 60% to this warming. Finally, we show that circulation changes caused by AA reductions can play an additional role in the redistribution of regional temperature changes apart from the radiative effects per se. Our results reveal a strong link between the recent acceleration of MED warming and global AA decreases, which unmask additional greenhouse gas-driven warming. This study highlights the sensitivity of the MED to global emission changes and the need for climate policies that couple air quality improvements with rapid greenhouse gas mitigation.

#### 25 1 Introduction

The Mediterranean Basin (MED) region is considered a hotspot of climate change (Cos et al., 2022; Lionello and Scarascia, 2018) that has experienced enhanced warming since the late 20<sup>th</sup> century, particularly during the summer (Doblas-Reyes et al., 2021; Gutiérrez et al., 2021). This trend cannot be fully explained by internal climate variability alone and can be attributed to external anthropogenic forcing (Allen et al., 2006; Hodnebrog et al., 2024; Rashid, 2021). The decrease of anthropogenic aerosols (AAs) over Europe, driven by European air pollution control policies (Turnock et al., 2016), has been

50

55

linked to temperature increases in the Euro-Mediterranean region (Boé et al., 2020; Dong et al., 2017; Glantz et al., 2022; Nabat et al., 2014, 2025; Urdiales-Flores et al., 2023; Zanis et al., 2012). However, other parts of the world such as North America (primarily the U.S. with the Clean Air Act) have also reduced their AA emissions, which could also influence the European climate (Elkins et al., 2025; Yang et al., 2020). Global concentrations of AAs in the atmosphere have decreased since around the 1980s, as suggested by changes in aerosol optical depth (AOD) over Europe and Eastern North America (Cherian and Quaas, 2020; Gulev et al., 2021; Klimont et al., 2013; Wei et al., 2019). Sulfate atmospheric loads have largely reduced during the last decades (Aas et al., 2019; Gulev et al., 2021), with subsequent radiative changes (Kalisoras et al., 2024; Myhre et al., 2017; Szopa et al., 2021).

Global AA decrease has caused changes in the Earth's radiative budget, which are expressed in terms of the effective radiative forcing (ERF) (Forster et al., 2021; Hansen et al., 2005; Sherwood et al., 2015). According to the Intergovernmental Panel on Climate Change (IPCC) Sixth Assessment Report (AR6) WGI (Forster et al., 2021; Gulev et al., 2021; Szopa et al., 2021), the ERF caused by AAs at the top-of-atmosphere (TOA) was negative throughout the industrial period over Europe, with a shift by the late 1970s – early 1980s towards less negative values and a weakening of the negative ERF in the first decades of the 21st century. Parallel to ERF weakening, an increase in shortwave radiation reaching the surface, a phenomenon referred to as the "brightening effect" (Wild, 2009, 2012), has been observed over Europe and the MED (Alexandri et al., 2017; Galanaki et al., 2021; MedECC, 2020; Nabat et al., 2014; Sanchez-Lorenzo et al., 2015; Wild, 2012). Arguably this can be attributed to reductions in sulfate aerosols (Allen et al., 2013; Nabat et al., 2014), which scatter incoming solar shortwave radiation and interact with clouds (Bellouin et al., 2020; Kasoar et al., 2016; Myhre et al., 2013).

A number of recent studies used observational datasets, reanalysis products, and global and regional climate models (GCMs and RCMs, respectively) to examine the enhanced surface warming and increase in temperature extremes over the Euro-Mediterranean region due to reductions in global AAs, which earlier had masked the continuous warming from greenhouse gases until around the 1980s (Boé et al., 2020; Cos et al., 2022; Dong et al., 2017; Drugé et al., 2021; Feng and Qian, 2024; Glantz et al., 2022; Hodnebrog et al., 2024; Nabat et al., 2014, 2025; Navarro et al., 2017; Samset et al., 2018; Tian et al., 2020; Urdiales-Flores et al., 2023). Decreases in AAs caused radiative imbalances over Europe and a shift from dimming to brightening, which was linked to increased warming at the surface (Folini and Wild, 2011; Glantz et al., 2022; Hodnebrog et al., 2024; Nabat et al., 2014, 2025; Sanchez-Lorenzo et al., 2015). Dong et al. (2017) showed that near-surface temperature and temperature extremes have increased since the 1990s in Western Europe during the summer, with AAs playing a key factor. Glantz et al. (2022) found an increase of about 1 °C attributed to AA decreases between the months April and September during the period 1979-2020 over Central and Eastern Europe. According to Nabat et al. (2025), each 0.1 reduction in AOD over Central Europe led to a 0.3 °C rise in near-surface temperature. In an earlier study, Nabat et al. (2014) showed that AA changes explain 81±16% of the brightening and 23±5% of the surface warming simulated for the period 1980-2012 over the European region based on reanalysis-driven coupled regional climate model. Turnock et al. (2016) estimated a mean surface temperature increase of around 0.45±0.11 °C over Europe between 1970 and 2010 from European AA reduction measures. Urdiales-Flores et al. (2023) carried out an attribution analysis to understand the drivers of accelerated warming

since the 1980s over the MED (based on observations, the median trend for 1981-2020 exceeded 0.4 °C per decade). They calculated a surface warming of 0.10 °C per decade over the MED for the period 1981–2020 attributed to AOD reductions, while they attributed 0.18 °C per decade to radiative forcing from well-mixed greenhouse gases and 0.03 °C per decade to soil moisture decrease. Boé et al. (2020) investigated summer temperature changes over Europe using global and regional climate models, and assessed the influence of AAs in surface warming with sensitivity experiments (also discussed in Boé, 2016). Yang et al. (2020) examined the trends of AAs over Europe during 1980-2018 and found a warming of 2.0 W m<sup>-2</sup> in Europe due to local decreases in sulfate loading, with 12% attributed to changes in non-European emissions. However, circulation changes attributed to reduced AA emissions also play an important role in near-surface temperature changes over Europe (Delworth et al., 2022; Dong and Sutton, 2021; Nabat et al., 2020; Zanis, 2009).

This study uses high-complexity, state-of-the-art Earth system models (ESMs) endorsed by the sixth phase of the Coupled Model Intercomparison Project (CMIP6) (Eyring et al., 2016) to investigate temperature changes over the Mediterranean Basin in terms of radiative and circulation changes throughout the historical period, focusing on two periods of interest: the decade that AAs peak over Europe (1970-1979) and the near-present decade (2005-2014). To our knowledge, this is the first study that implements two standard metrics (ERF and atmospheric radiative cooling; see Section 2.1) to quantify fast and total (i.e., including feedbacks) radiative responses resulting from global changes in AA emissions and tries to connect them with changes in temperature both at the surface and throughout the troposphere on annual and seasonal scales. Furthermore, AA-induced changes to atmospheric circulation are examined in an effort to explain temperature changes not related to AA radiative effects. Finally, observational datasets are used to evaluate the temperature changes over the MED simulated by ESMs on both annual and seasonal scales. The rest of the paper is structured as follows. Details about the CMIP6 ESMs and simulations used, along with a description of our methodology, are given in Sect. 2. Our results are discussed in Sect. 3, while in Sect. 4 the main conclusions of our work are summarized.

#### 2 Data and methodology

### 2.1 Data description

90

In this work, historical simulations from eight ESMs were used (Table 1), which were performed within the framework of the Aerosol Chemistry Model Intercomparison Project (AerChemMIP; Collins et al., 2017) and supported by CMIP6 (Eyring et al., 2016). Anthropogenic emissions of aerosols as well as precursors of aerosols and ozone (excluding methane) used by climate models are taken from van Marle et al. (2017) and Hoesly et al. (2018), while each model uses its own natural aerosol (e.g., dust and sea salt) emissions (Eyring et al., 2016). All models include both aerosol indirect effects (i.e., the cloud albedo effect and the cloud lifetime effect) (Albrecht, 1989; Twomey, 1974, 1977), except for CNRM-ESM2-1 which only represents the cloud albedo effect (and hence is lacking a cloud lifetime effect). Additionally, MRI-ESM2-0 and NorESM2-LM parameterize aerosol effect on ice clouds, thus impacting the LW ERF (e.g., Smith et al., 2020; Zelinka et al., 2023). A short description of each ESM is given in the Appendix.

Within the framework of CMIP6 (Eyring et al., 2016) and AerChemMIP (Collins et al., 2017) ESMs performed two sets of transient historical experiments for the period between 1850 and 2014 (Table 2). The first set includes the CMIP6 "historical" (Eyring et al., 2016) and "hist-piAer" (Collins et al., 2017) coupled-ocean experiments, and the second set includes the "histSST" and "histSST-piAer" experiments with prescribed sea surface temperatures (SSTs) and sea ice (Collins et al., 2017). The "historical" experiment branches from the "piControl" experiment and is forced by time-evolving, externally imposed anthropogenic and natural forcings that are based on observations, such as solar variability, volcanic aerosols, and changes in land use and greenhouse gas and aerosol emissions or concentrations (Eyring et al., 2016). The "histSST" experiment shares the same forcings as the "historical" and SSTs and sea ice are prescribed to the monthly mean time-evolving values from the corresponding "historical" simulation of each model. Including evolving SSTs, rather than using a fixed repeating climatology throughout the simulation, means that the underlying climate state in "histSST" is consistent with the "historical" experiment (Collins et al., 2017). The "hist-piAer" and "histSST-piAer" experiments are identical to "historical" and "histSST", respectively, but they use fixed pre-industrial anthropogenic aerosol and aerosol precursor emissions corresponding to the year 1850, while all other forcings remain the same (Collins et al., 2017).

To evaluate the near-surface air temperature evolution in CMIP6 coupled-ocean experiments, three gridded observational datasets (Schneider et al., 2013) were employed: (a) the Berkeley Earth (https://berkeleyearth.org/data/) dataset (Rohde and Hausfather, 2020), which provides high-resolution (1° x 1°) monthly global land-ocean temperature data starting from the year 1850, (b) the Climatic Research Unit of the University of East Anglia (https://crudata.uea.ac.uk/cru/data/hrg/) gridded Time Series (CRU TS v4.08) (Harris et al., 2020), which offers a high-resolution (0.5° x 0.5°) monthly grid of land-based observations going back to 1901, and (c) the University of Delaware land temperature monthly gridded data (v4.01) at high resolution (0.5° x 0.5°) beginning in 1900 and provided by the National Oceanic and Atmospheric Administration (NOAA) Physical Sciences Laboratory (PSL), Boulder, Colorado, USA, from their website (https://psl.noaa.gov). The CRU TS and University of Delaware datasets contained absolute temperature values (in °C), whereas Berkley Earth temperature anomalies relative to the 1951-1980 baseline climatology, which were converted to absolute temperature values (in °C).

## 2.2 Calculation methodology

In this work, monthly outputs from the ESM simulations were used (summarized in Table 3). All data were first regridded to a common spatial grid (2.8125° × 2.8125°) by applying bilinear interpolation. The effects of changes in global anthropogenic emissions of aerosols on AOD, near-surface air temperature (TAS), and circulation (sea level pressure, geopotential height, and winds) were calculated by subtracting "hist-piAer" from the "historical". Changes in AOD, TAS, and circulation are calculated using the coupled experiments.

The transient aerosol ERF was calculated over the historical period using the prescribed-SST experiments. By fixing SSTs and sea ice, all other parts of the system are allowed to respond until reaching steady state (Hansen et al., 2005). Consequently, ERF can be diagnosed as the difference in the net flux at the top-of-atmosphere (TOA) between the perturbed and the control experiment (Hansen et al., 2005; Sherwood et al., 2015). While not technically an ERF (since SSTs and sea ice

are evolving), the impact of transient SSTs and sea ice on ERF is regarded to be small (Collins et al., 2017; Forster et al., 2016). Here, ERF was estimated at the top-of-atmosphere (ERF<sub>TOA</sub>) and at the surface (ERF<sub>SURF</sub>) following Forster et al. (2016). ERF<sub>TOA</sub> was calculated as the difference between the incoming and outgoing radiative fluxes at the top-of-atmosphere (Eq. 1), while ERF<sub>SURF</sub> as the difference between the surface downwelling and upwelling radiative fluxes (Eq. 2):

$$ERF_{TOA} = \Delta F_{TOA} = \Delta [rsdt - (rsut + rlut)], \tag{1}$$

$$ERF_{SURF} = \Delta F_{SURF} = \Delta [(rsds + rlds) - (rsus + rlus)],$$
 (2)

where  $F_{TOA}$  and  $F_{SURF}$  are the net (downward minus upward) radiative fluxes at the top-of-atmosphere and at the surface, respectively, and  $\Delta$  denotes the difference between the perturbation and the control experiment ("histSST" minus "histSST-piAer").

While ERF is a metric of the initial perturbation caused by a forcing agent (since SSTs are prescribed), the atmospheric radiative cooling (ARC) is a standard metric that quantifies the radiative loss of the atmospheric column from its top and bottom boundaries (TOA and surface, respectively) after allowing parts of the Earth-atmosphere system to be adjusted by the ocean and sea ice response (since it is calculated from the coupled experiments here). ARC has been traditionally used in papers investigating precipitation changes via changes in the atmospheric energy budget (e.g., Dagan et al., 2019; Hodnebrog et al., 2016; Liu et al., 2018; Muller and O'Gorman, 2011; Naegele and Randall, 2019; Richardson et al., 2018; Tang et al., 2018; Zhang et al., 2021c), but here it is used supplementarily with ERF for two main reasons: (a) to investigate differences in the radiative budget between the prescribed-SST runs (corresponding to fast climate responses) and the coupled runs (representing the total climate responses), and (b) to link the total temperature changes with the total radiative changes (ARC) rather than just the fast responses in the radiative balance (ERF).

ARC is defined as the net shortwave (SW) and longwave (LW) radiation loss of the atmospheric column (Muller and O'Gorman, 2011; O'Gorman et al., 2012) and is positive when the atmosphere is radiatively cooled (i.e., loses energy) and negative when the atmosphere is radiatively warmed (i.e., gains energy). It is calculated as the sum of the net SW and LW radiative fluxes from the column (upwards at the top-of-atmosphere and downwards at the surface, i.e. positive when exiting the column) following Muller and O'Gorman (2011) (Eq. 3):

$$ARC = ARC_{TOA}^{NET} + ARC_{SURF}^{NET} = \Delta F_{TOA}^{NET} + \Delta F_{SURF}^{NET} = \Delta [(rsut + rlut) - rsdt] + \Delta [(rsds + rlds) - (rsus + rlus)], (3)$$

where  $F_{TOA}^{NET}$  and  $F_{SURF}^{NET}$  are the net (SW plus LW) radiative fluxes at the top-of-atmosphere and at the surface, respectively, and  $\Delta$  denotes the difference between the perturbation and the control experiment ("historical" minus "hist-piAer"). Based on Eq. (3), the SW and LW components of ARC<sub>TOA</sub> and ARC<sub>SURF</sub> were calculated as follows (Eq. 4-7, respectively):

$$ARC_{TOA}^{SW} = \Delta(F_{TOA}^{SW}) = \Delta(rsut - rsdt), \qquad (4)$$

$$ARC_{SURF}^{SW} = \Delta(F_{SURF}^{SW}) = \Delta(rsds - rsus), \qquad (5)$$

$$160 \quad ARC_{TOA}^{LW} = \Delta(F_{TOA}^{LW}) = \Delta(rlut) , \qquad (6)$$

$$ARC_{SURF}^{LW} = \Delta(F_{SURF}^{LW}) = \Delta(rlds - rlus), \qquad (7)$$

where  $F_{TOA}^{SW}$  and  $F_{SURF}^{SW}$  are the SW and  $F_{TOA}^{LW}$  and  $F_{SURF}^{LW}$  are the LW radiative fluxes at the top-of-atmosphere and at the surface, respectively, and  $\Delta$  is the same as in Eq. (3). Positive (negative) values of ARC<sub>SURF</sub> denote that the atmosphere is radiatively cooled (warmed) from the surface while the surface is radiative warmed (cooled) by the same amount of energy.

The statistical significance of each ESM's results was tested at the 95% confidence level using a two-tailed t-test. The robustness of the multi-model ensemble results was estimated using a comprehensive method based on statistical significance and the agreement between ESMs on the sign of change. In cases where at least 80% (i.e., at least 6 out of 8) of the ESMs show statistically significant change and at least 80% of the ESMs agree with the multi-model mean on the sign of change, then these results are considered robust. If less than 80% (i.e., 5 or less out of 8) of the ESMs show not statistically significant change or less than 80% of the models do not agree with the multi-model mean on the sign of change, then they are characterized as non-robust or conflicting, respectively. To study the effects caused by global AA reduction, two periods were selected: (a) 1970-1979, which corresponds to the peak aerosol decade over Europe, and (b) 2005-2014 as the last decade available in CMIP6 historical simulations which represents the near-present. When examining a single period (1970-1979 or 2005-2014), changes reflect the effect of AAs relative to the pre-industrial period, whereas the difference between the two periods (2005-2014 minus 1970-1979) represents the effects of AA decreases since around 1980. In the next section, the multi-model ensemble results are presented for the study region (10°W-40°E, 30°N-45°N). Area-weighted mean values refer to the multi-model means unless stated otherwise.

#### 3 Results

## 3.1 Radiative changes

As described earlier, AA concentrations peak in the late 1970s to early 1980s in Europe and the MED. Consequently, AOD change reaches a maximum during that period (Fig. 1, left column), which is stronger in magnitude during the boreal summer (JJA) than the winter (DJF). Likewise, transient ERF both at TOA and surface attained its most negative values during the late 1970s - early 1980s on an annual basis, dominated by the evolution of ERF in JJA (Fig. 1, middle column). In DJF both ERFs (ERF<sub>TOA</sub> and ERF<sub>SURF</sub>) are weaker in magnitude than in JJA resulting from less incoming radiation, with ERF<sub>SURF</sub> also being weaker than ERF<sub>TOA</sub> during 1975-1985. Inter-model variability is large in DJF, especially in the case of ERF, with values reaching up to around ±3 W m<sup>-2</sup> during the negative peak. Similarly, the net ARC<sub>SURF</sub> shows a trend towards more negative values until around 1980 followed by a reverse positive trend with a weakening in magnitude towards the end of the historical period on an annual scale (Fig. 1, right column), driven by the evolution of the SW ARC<sub>SURF</sub>. In DJF, the net ARC<sub>SURF</sub> and its SW and LW components show smaller changes throughout the historical period compared to ARC<sub>SURF</sub> during the summer time. The SW ARC<sub>SURF</sub> during the summer exhibits the most negative peak in the late 1970s – early 1980s, with values exceeding -10 W m<sup>-2</sup>. The LW component of ARC<sub>SURF</sub> is slightly positive (i.e., surface warming) and it evolves in an opposite

way to the negative SW component of ARC<sub>SURF</sub> (i.e., surface cooling). The SW ARC<sub>SURF</sub> practically drives the negative net ARC<sub>SURF</sub> (indicating surface cooling) and its temporal evolution. This can be explained by the evolution of the AOD of sulfate aerosols over the MED (Fig. S1 in the Supplement), which is the dominant AA type and closely follows the trend of total AOD (Fig. 1). However, interactions between aerosols and clouds could also play a role in SW ARC<sub>SURF</sub>.

High emissions of AAs and their precursors in the period 1970-1979 over Europe have led to an increase in AOD mainly in Central Europe and its surrounding regions relative to the pre-industrial period, thus increasing AOD in the northern parts of the MED (Fig. 2). The annual mean AOD change due to AAs during 1970-1979 over the MED is 0.11±0.04 (inter-model variability; 1 standard deviation), with changes being slightly larger (smaller) in magnitude in JJA (DJF). Changes in AA emissions and concentrations have led to reductions in AOD in 2005-2014 (with respect to 1970-1979) in Central European regions and consequently in the MED of -0.05±0.02 annually, with the largest AOD reductions shown in JJA (-0.07±0.02). Spatially, Southeastern Europe (Balkan Peninsula) shows the largest annual and seasonal AOD changes between the periods 1970-1979 and 2005-2014 within the MED region. The multi-model results are robust for most of the MED (except the southern and western parts) on both annual and seasonal scales. The MPI-ESM-1-2-HAM and MRI-ESM2-0 models show the largest AOD change between the two periods (-0.086 and -0.087, respectively; Table 4), with the former exhibiting the largest change in DJF (-0.115; Table 5) and the latter in JJA (-0.127; Table 6).

Changes in ERF<sub>TOA</sub> spatial pattern are presented in Fig. 3, while ERF<sub>SURF</sub> spatial patterns are illustrated in Fig. S2 (in the Supplement). The annual area-weighted means for each ESM and the multi-model ensemble over the MED are shown in Table 4. The respective seasonal means are presented in Tables 5 (for DJF) and 6 (for JJA). AAs exert a negative ERF<sub>TOA</sub> over Central Europe in 1970-1979 relative to the pre-industrial period on annual scale, which greatly increases in magnitude in JJA and spreads mainly over the eastern and partly over the western and northern parts of Europe, but weakens in DJF and relocates over the MED (Fig. 3). The largely negative  $ERF_{TOA}$  over regions north of the MED is offset by the positive  $ERF_{TOA}$  over the Sahara Desert, leading to a multi-model mean ERF<sub>TOA</sub> of -4.16±1.74 W m<sup>-2</sup> over the MED in JJA, which is weaker than the annual mean of -4.43±1.55 W m<sup>-2</sup>. ERF<sub>SURF</sub> exhibits a spatial pattern that is very similar to ERF<sub>TOA</sub>, but it is stronger in magnitude (i.e., more negative) due to the more negative ERF<sub>SURF</sub> over Africa (Fig. S2). The weakening of the forcing due to AA reduction leads to a smaller annual mean ERF<sub>TOA</sub> of -2.06±1.00 W m<sup>-2</sup> over MED in 2005-2014 (Fig. 3), which becomes even smaller in JJA (-0.99±1.70 W m<sup>-2</sup>) because of the weaker radiative forcing gradient between Central Europe and Northern Africa. However, the multi-model results are robust only over the Eastern MED on an annual scale and in DJF. The large intermodel variability in the summer during the period 2005-2014 stems from MRI-ESM2-0 (Table 4), which simulates a strongly negative ERF<sub>TOA</sub> (-4.77 W m<sup>-2</sup>). The MRI-ESM2-0 model produces the most negative ERF<sub>TOA</sub> and ERF<sub>SURF</sub> over the MED in all cases except in the winter during the period 2005-2014, where MPI-ESM-1-2-HAM takes the lead in both ERFs (Table 5). The largest changes in ERF<sub>TOA</sub> and ERF<sub>SURF</sub> between the two examined periods are seen for EC-Earth3-AerChem, MPI-ESM-1-2-HAM, MRI-ESM2-0, and UKESM1-0-LL, which differ in magnitude depending on the season. In general, as illustrated in Figs. 3 and S2, the ensemble differences of ERF<sub>TOA</sub> and ERF<sub>SURF</sub> between the 2005-2014 and the 1970-1979 periods indicate that all ESMs point towards an ERF weakening (less negative ERF) that is more prominent in JJA, which dominates the spatial

pattern of the annual change in ERF (mind that the differences in DJF are weaker and non-robust). The annual and summer ERF weakening is confined to the European part of the MED, where the multi-model results are robust.

Changes in the spatial pattern of net ARC<sub>SURF</sub> are shown in Fig. 4, while the SW and LW components of ARC<sub>SURF</sub> are presented in Figs. S3 and S4, respectively, in the Supplement. The net, SW, and LW ARC<sub>TOA</sub> patterns are presented in Figs. S5-S7, respectively, while the net, SW, and LW ARC of the entire atmospheric column are presented in Figs. S8-S10, respectively, in the Supplement. The net, SW, and LW ARC<sub>SURF</sub> values are given in Tables 4-6, while the rest of the ARC decomposition (ARC<sub>TOA</sub> and ARC for the entire atmospheric column) is presented in Tables S1-S3 in the Supplement.

Overall, the spatial pattern of net ARC<sub>SURF</sub> (Fig. 4) is very similar to those of ERF<sub>TOA</sub> (Fig. 3) and ERF<sub>SURF</sub> (Fig. S2) and is predominantly affected by the pattern of its SW component (Fig. S3) during both periods. The annual mean net ARC<sub>SURF</sub> in 1970-1979 is -5.49±1.46 W m<sup>-2</sup> over the MED and is largest over Central and Eastern Europe (Fig. 4). This negative net ARC<sub>SURF</sub> results from a larger SW radiative energy loss at the surface (surface cooling) and a smaller LW radiative gain at the surface (surface warming; Table 4). The annual spatial patterns of net and SW ARC<sub>SURF</sub> in the period 1970-1979 are dominated by their respective patterns in JJA. The highly negative net ARC<sub>SURF</sub> in the summer is spread all over the Euro-Mediterranean region and has a multi-model mean value of -7.60±2.30 W m<sup>-2</sup> over the MED. In the winter of 1970-1979, the net ARC<sub>SURF</sub> is weaker (-2.72±1.34 W m<sup>-2</sup>) and shows a robust increase over continental Europe after the AA decrease (surface warming; Fig. 4f) towards the period 2005-2014. On the other hand, the difference between the periods 2005-2014 and 1970-1979 in net ARC<sub>SURF</sub> during the summer is stronger in magnitude (2.71±1.26 W m<sup>-2</sup>), but it is not a robust feature among ESMs. It becomes clear that the SW component drives the pattern and evolution of the net ARC<sub>SURF</sub>, while the LW component partly masks the ARC change at the surface. The LW ARC<sub>SURF</sub> is positive during 1970-1979 both annually and seasonally, indicating radiative energy gain at the surface (or loss for the atmosphere), but weakens after the AA reduction (Fig. S4). At TOA the SW component (Fig. S6) still dominates the net ARC<sub>TOA</sub> (Fig. S5) but there are some qualitative differences to ARC<sub>SURF</sub>. During 1970-1979 the net ARC<sub>TOA</sub> is positive over the Euro-Mediterranean region (Fig. S5), indicating radiative loss for the atmosphere that stems from its SW component (Fig. S6), despite the LW atmospheric radiative gain at TOA partly masking the aforementioned loss (Fig. S7). The annual spatial pattern is still driven by the JJA pattern in both cases (SW and LW). After the AA decrease, there is a larger change in the SW (less SW atmospheric cooling at TOA in 2005-2014) that dominates the LW ARC changes (less LW atmospheric gain at TOA in 2005-2014) resulting in a reduced net ARC<sub>TOA</sub> (0.80±0.95 W m<sup>-1</sup> <sup>2</sup>) in the period 2005-2014 (Fig. S5). This change in net ARC<sub>TOA</sub> is more prominent over Central Europe during the summer months. Although in the transition from 1970-1979 to 2005-2014 the net atmospheric radiative gain from the surface weakens in JJA (i.e., summer surface warming; Table 6), the net atmospheric radiative loss at TOA becomes negative (i.e., atmospheric radiative gain), resulting in an energy flow into the atmospheric column (Table S3). Despite this qualitative change at TOA, the quantitative change at the surface is larger, resulting in a less negative mean ARC<sub>NET</sub> in the summer during the period 2005-2014, especially over the central and eastern parts of the MED (Fig. S8). The more negative annual mean ARC<sub>NET</sub> over the MED in 2005-2014 relative to 1970-1979 (Fig. S8) results from a larger decrease in the net atmospheric radiative loss at TOA (Table S1) compared to the smaller decrease in the net atmospheric radiative gain from the surface (Table 4). Decreases

in AAs cause the largest changes in SW ARC both at the surface and at TOA, mainly in the summer (Tables 6 and S3, respectively). Changes in the SW component, which are larger at TOA than at the surface, cause a stronger SW radiative gain for the entire atmospheric column (atmospheric warming) in 2005-2014 than in 1970-1979 (Fig. S9). On the other hand, changes in the LW component, which are also larger at TOA than at the surface, cause a weaker LW radiative gain for the entire atmosphere (atmospheric cooling) in 2005-2014 when compared to 1970-1979 (Fig. S10). The annual total SW change dominates the annual total LW change, thus resulting in stronger atmospheric warming over the MED region after the AA reductions (Fig. S8). However, in JJA the total LW change (Fig. S10) is larger than the SW change (Fig. S9), causing a weakening in summer atmospheric warming over the MED after the decrease in AAs (Fig. S8).

## 3.2 Temperature changes

The evolution of TAS in the MED throughout the historical period because of AAs (Fig. 5) follows the evolution of transient ERF and net ARC<sub>SURF</sub> (Fig. 1). The annual mean TAS exhibits a negative peak in the late 1970s - early 1980s, which is stronger in JJA compared to DJF, showing an increasing trend towards more positive values by the end of the historical period. To gain confidence in the AA-induced TAS changes presented in Fig. 5, the ability of CMIP6 ESMs to capture the historical trends of TAS in the MED is examined. The TAS evolution over land from the "historical" simulations is compared to those from three gridded observational datasets (Fig. 6). On annual scale and during the summer, observations and the "historical" simulation agree on the evolution of land TAS in MED, with a cooling trend after the 1940s that peaks in the mid-1970s and a warming trend afterwards. The absolute temperature levels in "historical" simulation are in good agreement with the University of Delaware and CRU TS datasets, especially in JJA, whereas there is a distinct warm bias in JJA and cold bias in DJF with respect to the Berkley Earth dataset. On an annual scale over land, the three observational datasets show a warming ranging from 1.14 °C to 1.20 °C between the periods 2005-2014 and 1970-1979, and the ensemble from the "historical" simulations indicate a relatively similar warming of 1.34 °C with a small overestimation in the range of 12-18% (Table 7). The ensemble from the "hist-piAer" simulations shows a smaller warming of 0.68 °C, thus indicating that AA reductions contribute roughly 0.66 °C (49%) for the warming in transient historical simulations between 2005-2014 and 1970-1979. In JJA over land, the three observational datasets show a stronger warming in the range of 1.68 to 1.81 °C between 2005-2014 and 1970-1979, and the ensemble from the "historical" simulations indicates a similar warming of 1.67 °C with a slight underestimation in the range of 1-8%. The ensemble from the "hist-piAer" simulations shows a smaller warming of 1.02 °C, which indicates that reductions in AAs contribute roughly 0.65 °C (39%) for the warming in CMIP6 historical simulations between the periods 2005-2014 and 1970-1979. In the winter, land air temperature evolves differently in the "historical" simulation compared to observations. While ESMs exhibit a similar, yet weaker cooling-warming pattern which is also seen in JJA, observational datasets show a general temperature increase since the 1970s. For example, in DJF over land, the three observational datasets show a slight warming in the range of 0.21 to 0.34 °C between 2005-2014 and 1970-1979, while the ensemble from the "historical" simulations indicates an overestimated warming of 1.19 °C, with "hist-piAer" simulations revealing that AA decrease contributes roughly 0.71 °C (60%) to this warming. CMIP6 models produce a distinct cooling during the period 1960-

1990 in DJF followed by an excessive warming trend from around 1995 onwards, which can arguably be attributed to aerosol concentration biases in the ESMs (Zhang et al., 2021b). In addition, the relatively weaker radiative effects from AAs in DJF than in JJA (as discussed in Section 3.1), and atmospheric circulation, which is generally more disturbed in boreal winter than in boreal summer in the mid-latitudes (Akritidis et al., 2021), influence the differences in DJF land temperature evolution between ESMs and observations. It has to be highlighted here that all three observational datasets used here show significant discrepancies between each other in representing the seasonal land TAS evolution, especially in DJF. In some cases these discrepancies are larger between the observational datasets than between the observational data and the CMIP6 "historical" ensemble.

Changes in the spatial pattern of TAS due to AAs as simulated by CMIP6 models are presented in Fig. 7. AAs cause a surface cooling relative to the pre-industrial era, which is most intense during 1970-1979 with an annual mean of -1.49±0.52 °C over the MED. The central and eastern parts of Europe experience the largest TAS decrease on both annual and seasonal scales. The surface cooling in the MED for the period 1970-1979 is more enhanced in the summer (-1.89±0.52 °C) than in the winter (-1.14±0.70 °C). The same applies to both Europe and Northern Africa, a feature that is robust among ESMs. After AA emission reduction, towards the period 2005-2014, mainly Central Europe and the Eastern MED undergo a surface warming both annually and seasonally (as can be noted in Fig. 7 from the difference between the periods 2005-2014 and 1970-1979). Specifically, in the MED the mean increase in TAS is of the same magnitude (around 0.7 °C) on an annual scale and during the winter and summer months. However, the annual change is robust in most of the MED, whereas in DJF it is not statistically significant (despite the agreement on the sign of change between ESMs). In JJA, the robust results are confined in the Eastern MED, which is highlighted as a hotspot of summer warming due to AA reduction in all ESMs used in this work. The largest surface warming from 1970-1979 to 2005-2014 is produced by the EC-Earth3-AerChem and the UKESM1-0-LL models on both annual and seasonal scales, while the smallest by the BCC-ESM1 model.

The warming caused by the decrease in AA emissions and concentrations is also apparent at higher levels in the troposphere. In Fig. 8 (9) the zonal (meridional) temperature means from the surface to 50 mb are presented considering only the MED longitudes (latitudes). The warming extends up to 200 hPa over the MED region (defined by the vertical dashed lines) as part of the Northern Hemisphere (NH) extratropical warming from 1970-1979 to 2005-2014 (Fig. 8). In the summer, the temperature increase affects a larger part of the NH troposphere, reaching up to 90 °N, whereas it is more geographically confined in the winter. The epicenter of the annual mean warming is located at the MED longitudes, with an exception in JJA, during which the warming extends to around 60 °E (Fig. 9). The magnitude of the warming decreases with height, except in the case of the NH midlatitudes in DJF (Fig. 8).

Based on the results presented above, AA-induced TAS spatial changes cannot be fully explained by radiative changes alone. ERF<sub>TOA</sub> and the net ARC<sub>SURF</sub> changes between the periods 1970-1979 and 2005-2014 are not always spatially collocated with the respective TAS changes in the Euro-Mediterranean region (20°N-65°N, 25°W-55°E), as revealed by their spatial correlation coefficients on an annual scale (0.65 and 0.60, respectively), in DJF (0.52 and 0.62, respectively), and in JJA (0.69 and 0.64, respectively). This reveals that apart from the radiative effects and aerosol-cloud interactions (Kalisoras et al., 2024),

atmospheric circulation changes induced by AAs may modify horizontal and vertical heat transport, thus playing an additional role in the redistribution of TAS changes. To examine the thermal transfer through atmospheric circulation due to AAs, circulation changes at 850 hPa and sea-level pressure (SLP) relative to the pre-industrial period and due to decreases in AAs after around 1980 are shown in Fig. 10 and Fig. S11, respectively.

On an annual scale, AA reduction (Fig. 10c and Fig. S11c) causes a cyclonic anomaly over the North Atlantic and Western Europe, which extends towards the central and eastern parts of Europe. This atmospheric circulation anomaly at 850 hPa and SLP is responsible for an enhancement of the westerly/southwesterly flow over the MED latitudes, along with an inflow from the northwestern parts of Africa towards Southern Europe. This indicates a warm advection anomaly which could contribute to the annual local TAS maximum over Southeastern Europe and Eastern MED (Fig. 7c). During the winter, an anticyclonic anomaly (relative to the pre-industrial era) over Scandinavia in 1970-1979 (Fig. 10d), which could induce a cold advection anomaly towards Central and Southeastern Europe, strongly weakens or even disappears in the period 2005-2014 (Fig. 10e; also seen in Fig. S11e). Consequently, after the decrease in AAs from 1970-1979 to 2005-2014, a cyclonic anomaly extending from the North Atlantic to Eastern/Northeastern Europe (Fig. 10f and Fig. S10f) intensifies the westerly/southwesterly flow over the European midlatitudes (as well as the European part of the MED) and could possibly contribute to the warming pattern over Central and Eastern Europe (Fig. 7f), along with the warming due to radiative effects. On the other hand, in JJA there are cyclonic anomalies (relative to the pre-industrial period) caused by AAs that extend from the North Atlantic to Northwestern Africa and Southeastern Europe during both 1970-1979 and 2005-2014. These anomalies could impose a southerly warm advection over the Eastern MED that may partly mask the radiative cooling effect of AAs in the region for both periods (Fig. 10g and Fig. 10h, respectively). The circulation anomalies, resulting from the difference between 2005-2014 and 1970-1979 following the reduction of AAs, are generally weak (Fig. 10i). It could be mentioned that the emergence of a non-robust cyclonic anomaly over the Eastern MED and the Middle East (Fig. 10i and Fig. S11i) may contribute to the summer warming amplification over the Eastern MED possibly through a weakening of the Etesian wind system (Dafka et al., 2016) and a warm advection anomaly from Northeastern Africa. However, it should be noted that ESMs show large differences when simulating circulation changes attributed to AA reduction, as the large inter-model variability and the lack of statistical significance suggest.

# 4 Conclusions

In this work, changes to temperature and the radiative budget attributed to changes in AA emissions globally were examined for the Mediterranean Basin. Model output from the CMIP6 historical simulations with all forcings varying and from the AerChemMIP simulation with fixed pre-industrial AA and aerosol precursor emissions produced by eight CMIP6 Earth system models were analyzed. Annual and seasonal changes in AOD, ERF, ARC, temperature, and circulation were quantified for two periods: the peak aerosol period and the near-present period, which represent AA-induced changes relative

to the pre-industrial era. Although a clean attribution could not be performed, we assess changes between these two periods to broadly result from reductions in AAs driven by air pollution control policies.

Reductions in AA emissions due to air pollution control policies cause a decrease in AOD over the MED, in line with the findings of other papers. The decrease in the overall AOD is mainly affected by the sulfate emissions decrease and, thus, sulfates can be considered the main driver in AOD change. The ERF negative peak coincides with the positive AOD peak around the 1980s over the Euro-Mediterranean region, and after that follows a positive trend until the second decade of the 21st century. The magnitude of this change is robust over the northern half of the MED and is largest at TOA than the surface. The JJA pattern of ERF dominates its annual spatial pattern in all cases.

Changes in the net ARC<sub>SURF</sub> spatial pattern are dominated by its SW component on both annual and seasonal scales. The midlatitudes of Europe undergo the largest and most robust change in ARC<sub>SURF</sub>, with JJA changes playing a pivotal role in the changes seen on an annual scale. The AA-induced changes in the SW component of ARC<sub>SURF</sub> are significantly larger in magnitude than in the LW. The total SW ARC changes due to AA reduction are of opposite signs at TOA (negative) and the surface (positive). This means that after the removal of AAs the atmosphere loses a smaller amount of energy at TOA and simultaneously gains less energy from the surface both annually and seasonally. As the TOA SW change is larger, in absolute values, than the surface SW change, the entire atmospheric column results in gaining more SW radiation after the decrease in AAs. Overall, the more negative annual net ARC over the MED region after the removal of AAs results from a larger reduction in the net atmospheric radiative loss at TOA compared to the smaller decrease in the net atmospheric radiative gain from the surface.

The evolution of TAS is similar to that of ERF and the net ARC<sub>SURF</sub>, with a surface warming occurring around the late 1970s. This temperature increase is strongest during the summer months and is robust over the central and eastern parts of the MED. When compared to observational data, CMIP6 models are found capable of capturing the enhanced MED surface warming due to AA reductions and generally agree on the temperature range with observations. From 1970-1979 towards the recent decade 2005-2014, the ensemble of CMIP6 simulations indicate a warming of 1.34 °C over land in the MED region on an annual scale and 1.67 °C during the summer, in relatively good agreement with three observational datasets. The model simulations also indicate that AA reductions between the two decades contribute to this warming roughly by 0.66 °C (49%) on an annual scale and 0.65 °C (39%) in the summer. In the winter, land air temperature evolves differently in the "historical" simulations in comparison to observations. While ESMs exhibit a weak cooling-warming pattern, observational datasets show a general monotonic temperature increase since the 1970s. Furthermore, the ensemble of the "historical" model simulations indicates strongly overestimated warming for DJF (from 1970-1979 towards the recent decade 2005-2014) in comparison to the observational datasets. This could arguably be attributed to overestimated aerosol concentrations in ESMs responsible for the excessive cooling during the period 1960-1990 as well as the different climate model responses to the relatively weaker AA radiative effects and more disturbed mid-latitude atmospheric circulation conditions in DJF than in JJA. A warming due to AA decrease is also apparent at higher levels in the troposphere over the MED region, up to 200 hPa.

ERF<sub>TOA</sub> and the net ARC<sub>SURF</sub> changes (from 1970-1979 towards the recent decade 2005-2014) are not always spatially collocated with the respective TAS changes in the Euro-Mediterranean region, indicating that apart from the radiative effects and aerosol-cloud interactions, atmospheric circulation changes induced by AAs may modify horizontal and vertical heat transport, thus playing an additional role in the redistribution of regional TAS changes and being an important factor in the MED warming. Here, we find that cyclonic anomalies are induced by the AA decrease both at sea-level and at 850 hPa on an annual scale and during the boreal winter. These anomalies extend from the North Atlantic to Central and Eastern Europe and could be responsible for warm advection from Northern Africa. During the summer, there is a southerly warm advection towards Southeastern Europe from the northern parts of Africa, which may slightly mitigate the radiative cooling from AAs relative to the pre-industrial era. When AAs are decreased, JJA cyclonic anomalies emerge over the Eastern MED that could contribute to the amplification of the warming over the Eastern MED and Southeastern Europe through weakening of the Etesian wind system and warm advection anomaly from Northeastern Africa. Nevertheless, ESMs show large differences when simulating circulation changes attributed to AA reduction, as revealed by the large inter-model variability and the lack of statistical significance.

The enhanced warming over the Euro-Mediterranean region is an observed phenomenon that is also captured adequately by ESMs on an annual scale and during the summer months. However, due to the complex nature of aerosol-radiation-cloud interactions resolved in ESMs, which affect the amount of radiation reaching the surface and being trapped within the atmosphere, the contribution of AAs alone in the MED surface warming demands further work, especially when considering the lack of model agreement for the AA-induced changes on atmospheric circulation (convection and advection) that impact heat transfer and temperature anomalies. Parameterizations of climate models limit their ability to simulate certain climate feedbacks, sometimes resulting in large discrepancies between different ESMs. As such, further investigation is needed to robustly quantify the effects of AA reduction in the European and Mediterranean climate.

#### **Appendix**

In this section, the eight CMIP6 ESMs used in this work are described, focusing on their atmospheric general circulation and atmospheric chemistry models.

The Beijing Climate Center Earth System Model version 1 (BCC-ESM1) (Wu et al., 2020; Zhang et al., 2021a) uses the BCC Atmospheric General Model version 3 (BCC-AGCM3-Chem) (Wu et al., 2019) with interactive atmospheric chemistry. The 63 out of 66 chemical species considered in BCC-AGCM3-Chem are the same as those in the "standard version" of the Model for Ozone and Related chemical Tracers version 2 (MOZART2) (Horowitz et al., 2003). Sulfates are considered as prognostic and their treatment follows the Community Atmosphere Model (CAM) version 4 interactive atmospheric chemistry scheme (CAM4-chem) (Lamarque et al., 2012). Two types of organic and black carbon are treated in BCC-AGCM3-Chem, one water-insoluble and one water-soluble tracer. Sea salt and dust aerosols are classified into four size bins each. The mass mixing ratios of bulk aerosols are treated as prognostic variables following the NCAR Community Atmosphere Model (CAM3)

(Collins et al., 2004), directly affecting the radiative transfer in the atmosphere. BCC-AGCM3-Chem uses prognostic aerosol masses to estimate the liquid cloud droplet number concentration and considers indirect effects of aerosols (Wu et al., 2019).

The CNRM-ESM2-1 model (Michou et al., 2020; Séférian et al., 2019) developed by CNRM-CERFACS incorporates the Reactive Processes Ruling the Ozone Budget in the Stratosphere Version 2 (REPRO-BUS-C\_v2) atmospheric chemistry scheme for 63 chemistry species, in which chemical evolution is calculated only above the 560 hPa level (Michou et al., 2011; Morgenstern et al., 2017). Below that level, the concentrations of certain species are relaxed either toward the yearly evolving global mean abundances (Meinshausen et al., 2017) and for other species toward the 560 hPa value. The Tropospheric Aerosols for ClimaTe In CNRM (TACTIC\_v2) interactive tropospheric aerosol scheme has been adapted for use in CNRM-ESM2-1 (Michou et al., 2015), which implements a sectional representation of sea-salt and desert dust aerosols (3 bins each), black carbon and organic matter (2 bins each to distinguish hydrophilic and hydrophobic particles), and sulfates (1 bin). Sulfate precursors evolve in sulfate aerosols with dependence on latitude. These 12 species of tropospheric aerosols are considered as prognostic variables and are affected by transport, dry deposition, and sub-cloud and in-cloud scavenging (Séférian et al., 2019). The cloud droplet number concentration is determined on the concentrations of sea salt, sulfate aerosols, and organic matter as in Menon et al. (2002), thus representing only the cloud albedo effect.

The EC-Earth3-AerChem model (van Noije et al., 2021) is an extended version of EC-Earth3 (Döscher et al., 2022) with an added component to simulate tropospheric aerosols, methane, ozone, and atmospheric chemistry. It uses the McRad radiation package to treat the radiative transfer in clouds (Morcrette et al., 2008). Cloud droplet formation is described by the Abdul-Razzak and Ghan (2000) aerosol activation scheme. Atmospheric chemistry and aerosols are simulated with the Tracer Model version 5 release 3.0 (TM5-mp 3.0), which includes sulfate, black carbon, organic aerosols, mineral dust, and sea salt described by the modal aerosol microphysical scheme M7 (Vignati et al., 2004). Four water-soluble modes (nucleation, Aitken, accumulation, and coarse) and three insoluble modes (Aitken, accumulation, and coarse) are considered in M7, with a lognormal size distribution that has a fixed geometric standard deviation describing each mode. M7 describes the evolution of the total mass and total particle number of each species for each mode, and explains water uptake, new particle formation, and aging through coalescence and condensation (Vignati et al., 2004).

The GFDL-ESM4 model (Dunne et al., 2020; Horowitz et al., 2020) consists of the Geophysical Fluid Dynamics Laboratory (GFDL) Atmosphere Model version 4.1 (AM4.1), which includes an interactive tropospheric and stratospheric gas-phase and aerosol chemistry scheme. It is similar to its previous version (AM4.0) (Zhao et al., 2018a, b), but with some updates: it treats nitrate and ammonium aerosols explicitly, the aging of black and organic carbon aerosols from hydrophobic to hydrophilic forms depends on the calculated concentrations of hydroxyl radical (Liu et al., 2011), and the oxidation of sulfur dioxide and dimethyl sulfide to produce sulfate aerosols is driven by the gas-phase oxidant concentrations and cloud pH simulated by the online chemistry scheme (Paulot et al., 2016). The AM4.1 has an online representation of gas-phase tropospheric and stratospheric chemistry (Horowitz et al., 2020). Aerosols are represented as bulk concentrations of nitrate, ammonium, sulfate, and hydrophilic and hydrophobic black and organic carbon particles, plus five bins each for sea salt and

mineral dust aerosols. Sulfate and hydrophilic black carbon aerosols are regarded as being internally mixed by the radiation code (Larry W. Horowitz, personal communication, 2024).

The MPI-ESM-1-2-HAM model consists of the Max Planck Institute for Meteorology Earth System Model (MPI-ESM1.2), which uses the atmospheric general circulation model ECHAM6.3 (Mauritsen et al., 2019), coupled with the Hamburg Aerosol Model version 2.3 (HAM2.3) (Tegen et al., 2019). The ECHAM6.3–HAM2.3 (Lohmann and Neubauer, 2018; Mauritsen et al., 2019; Neubauer et al., 2019e; Tegen et al., 2019) uses a two-moment cloud microphysics scheme to examine aerosol-cloud interactions and to improve cloud simulation (Lohmann and Neubauer, 2018). Aerosol–cloud interactions are simulated in liquid, mixed-phase, and ice clouds (Neubauer et al., 2019e). In ECHAM6.3 the radiative effects of anthropogenic aerosols in the radiation calculation are treated using the MACv2-SP parameterization (Fiedler et al., 2017; Stevens et al., 2017). The aerosol microphysics module HAM (Stier et al., 2005; Zhang et al., 2012) calculates the evolution of aerosol particles considering the black and organic carbon, sulfate, sea salt, and mineral dust. In its default version, HAM simulates the aerosol size spectrum using the M7 (Vignati et al., 2004) scheme (Tegen et al., 2019). The updated version (HAM2.3) uses the scheme by Abdul-Razzak and Ghan (2000) for aerosol activation, along with other modifications for aerosol-cloud interactions, and updated emissions of aerosols and aerosol precursors from anthropogenic and natural sources (Neubauer et al., 2019e; Tegen et al., 2019).

The MRI-ESM2.0 model (Kawai et al., 2019; Oshima et al., 2020; Yukimoto et al., 2019c) consists of the atmospheric general circulation model with land processes MRI-AGCM3.5, coupled with the MRI Chemistry Climate Model version 2.1 (MRI-CCM2.1) atmospheric chemistry model, and the Model of Aerosol Species in the Global Atmosphere mark-2 revision 4-climate (MASINGAR mk-2r4c) aerosol model (Kawai et al., 2019; Yukimoto et al., 2019c). The MRI-CCM2.1 model simulates the evolution and distribution of ozone and other trace gases in the troposphere and middle atmosphere (Yukimoto et al., 2019c). The MASINGAR mk-2r4c model treats physical and chemical processes of atmospheric aerosols and includes black and organic carbon, non-sea-salt sulfate, mineral dust, sea salt, and aerosol precursor gases. Sea salt and mineral dust size distributions are divided into 10 discrete bins, whereas lognormal size distributions represent the sizes of the other aerosols (Oshima et al., 2020). External mixing is assumed for all aerosol species in this model, except for the radiation process in MRI-AGCM3.5, in which hydrophilic BC is assumed to be internally mixed with sulfate (Oshima et al., 2020; Yukimoto et al., 2019c). The conversion rate of hydrophobic to hydrophilic black carbon is based on the Oshima and Koike (2013) parameterization and depends on the rate at which condensable materials cover hydrophobic black carbon (Oshima et al., 2020). The aerosol activation into cloud droplets is based on the Abdul-Razzak et al. (1998), Abdul-Razzak and Ghan (2000), and Takemura et al. (2005) parameterizations.

The NorESM2-LM model (Seland et al., 2020) is the "low-resolution" version of the Norwegian Earth System Model version 2 (NorESM2), which is based on the Community Earth System Model (CESM2.1) (Danabasoglu et al., 2020) but with several modifications. NorESM2 employs the CAM6-Nor atmosphere model, which uses modified (compared to CESM2.1) parameterization schemes for aerosols and aerosol–radiation–cloud interactions (Kirkevåg et al., 2013, 2018), and the OsloAero6 atmospheric aerosol module (Kirkevåg et al., 2018), which describes the formation and evolution of sulfates, black

carbon, organic matter, mineral dust, sea salt, and secondary organic aerosols. The major difference between OsloAero6 and other aerosol modules is the division of tracers into "background" and "process" tracers. The "background" tracers are mainly primary emitted particles that form lognormal modes and contribute to the aerosol number concentration. The "process" tracers change the chemical composition and shape of the initially lognormal background modes (Kirkevåg et al., 2018).

The UKESM1 model (Mulcahy et al., 2020; Sellar et al., 2019, 2020), with HadGEM3-GC3.1 (Kuhlbrodt et al., 2018; Williams et al., 2018) as its physical core, uses the UK Chemistry and Aerosol (UKCA) interactive stratosphere-troposphere chemistry scheme (UKCA StratTrop) (Archibald et al., 2020) and the GLOMAP microphysical aerosol scheme (Mann et al., 2010, 2012). The UKCA StratTrop model employs 84 chemical species and represents the chemistry of 81 of these (oxygen, nitrogen, and carbon dioxide are not treated as chemically active species) (Archibald et al., 2020). GLOMAP is a two-moment modal aerosol microphysics scheme which treats the evolution, the sources and sinks of sulfates, black carbon, organic matter, and sea salt across five lognormal size modes. Five aerosol size modes are represented in GLOMAP (nucleation, soluble Aitken, insoluble Aitken, accumulation, and coarse). Mineral dust, on the other hand, is simulated independently using the CLASSIC dust scheme (Bellouin et al., 2011) and can be regarded to be externally mixed with the GLOMAP aerosols (Mulcahy et al., 2020).

Data availability. All data from the Earth system models used in this paper are available on the Earth System Grid Federation website and can be downloaded from there (https://esgf-node.llnl.gov/search/cmip6/, last access: 31 March 2025, ESGF, 2023). The Berkeley Earth dataset was downloaded from https://berkeleyearth.org/data/. Data for CRU TS v4.08 are available at https://data.ceda.ac.uk/badc/cru/data/cru\_ts/cru\_ts\_4.08/data. The University of Delaware dataset was retrieved from https://psl.noaa.gov/data/gridded/data.UDel AirT Precip.html.

Author contributions. AK and PZ conceptualized this study. AK, PZ, AKG, and DA designed the analysis. AK performed the formal analysis, produced the figures, and prepared the original draft. All authors contributed to the revision and editing of the paper.

Competing interests. The authors declare no conflicts of interest.

Acknowledgements. The authors acknowledge the World Climate Research Program, which promoted and coordinated CMIP6
through its Working Group on Coupled Modelling. The authors thank the climate modelling groups (listed in Table 1) for producing and making available their model output, the Earth System Grid Federation (ESGF) for archiving the data and providing access, and the multiple funding agencies who support CMIP6 and ESGF. The authors also acknowledge the NSF NCAR Climate Data Guide project for providing information and links to the observation datasets, the NOAA PSL, Boulder, Colorado, USA for providing the University of Delaware Terrestrial Temperature data and Kenji Matsuura for providing guidance, Dr. Kevin Cowtan and the Berkley Earth team for providing their global temperature data. The results presented in this study have been produced using the Aristotle University of Thessaloniki (AUTH) high-performance computing infrastructure and resources (https://hpc.it.auth.gr/).

Financial support. AK, DA, and PZ acknowledge support from the REINFORCE (improvements in the simulation of aerosol-climate linkages in earth system models: from global to Regional scalEs) research project; the research project is implemented in the framework of H.F.R.I call "Basic research Financing (Horizontal support of all Sciences)" under the National Recovery and Resilience Plan "Greece 2.0" funded by the European Union – NextGenerationEU (H.F.R.I. Project Number: 15155). AKG was partly supported by the EMME-CARE project, which received funding from the European Union's Horizon 2020 Research and Innovation Programme under grant agreement no. 856612, as well as matching co-funding by the Government of the Republic of Cyprus.

Table 1. Information on model resolution (horizontal and vertical), variant label, and references for each ESM used in this study. Each experiment (see Table 2) has a variant label r<sub>a</sub>i<sub>b</sub>p<sub>c</sub>f<sub>d</sub>, where a is the realization index, b is the initialization index, c is the physics index, and d is the forcing index.

| Model                 | Resolution<br>(longitude/latitude) | Vertical<br>Levels                  | Variant Label<br>(histSST/histSST-<br>piAer) | Variant Label<br>(historical/hist-<br>piAer) | Model References                                                                                             | Experiment<br>References                                             |
|-----------------------|------------------------------------|-------------------------------------|----------------------------------------------|----------------------------------------------|--------------------------------------------------------------------------------------------------------------|----------------------------------------------------------------------|
| BCC-ESM1              | 2.8125° x 2.8125°                  | 26 levels<br>Top Level:<br>2.19 hPa | -                                            | rlilplfl                                     | Wu et al. (2020);<br>Zhang et al. (2021a)                                                                    | (Zhang et al., 2018, 2020)                                           |
| CNRM-ESM2-1           | 1.4° x 1.4°                        | 91 levels<br>Top Level:<br>78.4 Km  | rlilp1f2ª                                    | rlilp1f2                                     | Michou et al. (2020);Séférian et al. (2019)                                                                  | (Seferian,<br>2018,<br>2019a, b,<br>c)                               |
| EC-Earth3-<br>AerChem | 0.7° x 0.7°b                       | 91 levels<br>Top Level:<br>0.01 hPa | rlilplfl                                     | rlilplfl                                     | Döscher et al. (2022);<br>van Noije et al. (2021)                                                            | (EC-Earth<br>Consortiu<br>m (EC-<br>Earth),<br>2020a, c, b,<br>2021) |
| GFDL-ESM4             | 1.25° x 1°°                        | 49 levels<br>Top Level:<br>0.01 hPa | rlilplfl                                     | rlilplfl                                     | (Dunne et al., 2020;<br>Horowitz et al., 2020)                                                               | (Horowitz<br>et al.,<br>2018a, b, c;<br>Krasting et<br>al., 2018)    |
| MPI-ESM-1-2-<br>HAM   | 1.875° x 1.875°                    | 47 levels<br>Top Level:<br>0.01 hPa | rlilplfl                                     | rlilplfl                                     | (Lohmann and<br>Neubauer, 2018;<br>Mauritsen et al., 2019;<br>Neubauer et al., 2019e;<br>Tegen et al., 2019) | (Neubauer<br>et al.,<br>2019a, b, c,<br>d)                           |
| MRI-ESM2-0            | 1.125° x 1.125°d                   | 80 levels<br>Top Level:<br>0.01 hPa | rlilplfl                                     | rlilplfl                                     | (Kawai et al., 2019;<br>Oshima et al., 2020;<br>Yukimoto et al.,<br>2019c)                                   | (Yukimoto<br>et al.,<br>2019a, b,<br>2020a, b)                       |
| NorESM2-LM            | 2.5° x 1.875°                      | 32 levels<br>Top Level:<br>3 hPa    | rlilp2fl°                                    | rlilplfl                                     | (Kirkevåg et al., 2018;<br>Seland et al., 2020)                                                              | (Oliviè et<br>al., 2019a,<br>b, c;<br>Seland et<br>al., 2019)        |
| UKESM1-0-LL           | 1.875° x 1.25°                     | 85 levels<br>Top Level:<br>85 km    | rlilp1f2                                     | rlilp1f2                                     | (Archibald et al.,<br>2020; Mulcahy et al.,<br>2020; Sellar et al.,<br>2019, 2020)                           | (O'Connor<br>, 2019a, b,<br>c; Tang et<br>al., 2019)                 |

<sup>&</sup>lt;sup>a</sup> The hist-piAer simulation is identical to the hist-piNTCF simulation as CNRM-ESM2-1 has no tropospheric chemistry, and therefore no ozone precursors, which means that the two configurations (hist-piAer and hist-piNTCF) are identical.

<sup>&</sup>lt;sup>b</sup> The 0.7° x 0.7° is approximate for the TL255 grid of IFS. Aerosols and atmospheric chemistry are simulated with the Tracer Model version 5 (TM5) with horizontal resolution 3° x 2° (lon x lat), with 34 levels in the vertical and a model top at about 0.1 hPa.

 $<sup>^{\</sup>circ}$  GFDL-ESM4 uses a cubed-sphere grid with  $\sim$ 100 km resolution (10 nominal horizontal resolution). Results are regridded to a 1.25 $^{\circ}$  x 1 $^{\circ}$  (lon x lat) grid for analysis.

<sup>d</sup> The 1.125° x 1.125° is approximate for the MRI-AGCM3.5. Aerosols are simulated with the Model of Aerosol Species in the Global Atmosphere mark-2 revision 4-climate (MASINGAR mk-2r4c) with horizontal resolution 1.875° x 1.875° (lon x lat), 80 vertical levels and a model top at 0.01 hPa.

of The distinction between p1 and p2 exists only for prescribed-SST simulations. These versions differ slightly in the treatment of the fluxes of energy, moisture, and momentum at the surface between the ocean and the atmosphere. The treatment of these fluxes in the p2 prescribed-SST simulations is closest to the fully-coupled (including active ocean) simulations. The p1 prescribed-SST simulations have a slightly different formulation of these fluxes (the flux calculation here corresponds more with the standard flux calculation in CESM2). Hence, the p2 version is closest related to the fully-coupled simulation (Dirk Olivié, personal communication, 2023).

Table 2. List of transient historical simulations covering the historical period (1850–2014) used in this work. The "histSST" and "histSST-piAer" experiments use prescribed SSTs and sea ice, whereas in the "historical" and "hist-piAer" experiments SSTs are allowed to evolve. The year indicates that the emissions or concentrations are fixed to that year, while "Hist" means that the concentrations or emissions evolve as for the "historical" experiment (more information in Collins et al., 2004 and Eyring et al., 2016).

| Experiment        | Type                             | CH <sub>4</sub> | N <sub>2</sub> O | Aerosol precursors | Ozone precursors | CFC/HCFC | MIP        |
|-------------------|----------------------------------|-----------------|------------------|--------------------|------------------|----------|------------|
| historical        | Coupled-<br>ocean<br>simulation  | Hist            | Hist             | Hist               | Hist             | Hist     | CMIP6      |
| hist-piAer        | Coupled-<br>ocean<br>simulation  | Hist            | Hist             | 1850               | Hist             | Hist     | AerChemMIP |
| histSST           | Prescribed-<br>SST<br>simulation | Hist            | Hist             | Hist               | Hist             | Hist     | AerChemMIP |
| histSST-<br>piAer | Prescribed-<br>SST<br>simulation | Hist            | Hist             | 1850               | Hist             | Hist     | AerChemMIP |

**Table 3.** Description of the CMIP6 variables used in this study.

| Variable  | Description                                       | Units               |
|-----------|---------------------------------------------------|---------------------|
| od550aer  | Ambient aerosol optical thickness at 550 nm       | Unitless            |
| od550bc   | Black carbon optical thickness at 550 nm          | Unitless            |
| od550dust | Dust optical thickness at 550 nm                  | Unitless            |
| od550oa   | Total organic aerosol optical thickness at 550 nm | Unitless            |
| od550so4  | Sulfate aerosol optical thickness at 550 nm       | Unitless            |
| psl       | Sea-level pressure                                | Pa                  |
| rlds      | Surface downwelling longwave radiation            | $W m^{-2}$          |
| rlus      | Surface upwelling longwave radiation              | W m <sup>-2</sup>   |
| rlut      | Top-of-atmosphere outgoing longwave radiation     | W m <sup>-2</sup>   |
| rsds      | Surface downwelling shortwave radiation           | W m <sup>-2</sup>   |
| rsdt      | Top-of-atmosphere incident shortwave radiation    | W m <sup>-2</sup>   |
| rsus      | Surface upwelling shortwave radiation             | W m <sup>-2</sup>   |
| rsut      | Top-of-atmosphere outgoing shortwave radiation    | W m <sup>-2</sup>   |
| ta        | Air temperature                                   | K (converted to °C) |
| tas       | Near-surface air temperature                      | K (converted to °C) |
| ua        | Eastward wind                                     | m s <sup>-1</sup>   |
| uas       | Eastward near-surface Wind                        | m s <sup>-1</sup>   |
| va        | Northward wind                                    | m s <sup>-1</sup>   |
| vas       | Northward near-surface Wind                       | m s <sup>-1</sup>   |
| zg        | Geopotential height                               | m                   |

© Author(s) 2025. CC BY 4.0 License.

Table 4. Annual mean values of AOD change (ΔAOD; unitless), ERF at TOA and at the surface (in W m<sup>-2</sup>), shortwave (SW), longwave (LW), and net (SW+LW) atmospheric radiative cooling at the surface (ARC<sub>SURF</sub>; in W m<sup>-2</sup>), and near-surface air temperature (TAS; in °C) 600 over the Mediterranean for the periods 1970-1979, 2005-2014 and their difference (2005-2014 minus 1970-1979). Values are presented for each model, along with the multi-model ensemble mean and the inter-model variability (one standard deviation; SD).

|                   | 1970-1979 |       |       |       |                 |       |         |       | 2005-2014 |       |       |                 |       |       |        |      | Diff. |      |           |      |      |  |
|-------------------|-----------|-------|-------|-------|-----------------|-------|---------|-------|-----------|-------|-------|-----------------|-------|-------|--------|------|-------|------|-----------|------|------|--|
| Model             | AOD       | ERF   |       | Α     | $ARC_{SURF} \\$ |       | TAC     | AOD   |           | RF    | A     | $ARC_{SURF} \\$ |       | TAS   | AOD    | ERF  |       | A    | $RC_{SU}$ | RF   | TAS  |  |
|                   | AOD       |       | SURF  | SW    | LW              | NET   | TAS AOD |       | SURF      | SW    | LW    | NET             | IAS   | AOD   | TOA    | SURF | SW    | LW   | NET       | IAS  |      |  |
| BCC-ESM1          | 0.160     | -     | -     | -8.01 | 1.67            | -6.33 | -1.55   | 0.115 | -         | -     | -5.87 | 1.14            | -4.73 | -1.34 | -0.045 | -    | -     | 2.14 | -0.54     | 1.60 | 0.21 |  |
| CNRM-ESM2-1       | 0.056     | -2.95 | -4.06 | -5.02 | 1.55            | -3.47 | -0.72   | 0.027 | -1.37     | -2.14 | -3.61 | 1.33            | -2.28 | -0.22 | -0.029 | 1.58 | 1.93  | 1.41 | -0.22     | 1.19 | 0.50 |  |
| EC-Earth3-AerChem | 0.101     | -4.74 | -5.88 | -9.21 | 2.83            | -6.38 | -2.55   | 0.056 | -1.68     | -4.26 | -7.78 | 3.19            | -4.58 | -1.19 | -0.045 | 3.06 | 1.63  | 1.43 | 0.36      | 1.79 | 1.36 |  |
| GFDL-ESM4         | 0.100     | -2.95 | -4.79 | -8.06 | 2.78            | -5.28 | -1.12   | 0.067 | -2.37     | -4.10 | -6.06 | 1.82            | -4.24 | -0.61 | -0.033 | 0.58 | 0.69  | 2.01 | -0.96     | 1.05 | 0.51 |  |
| MPI-ESM-1-2-HAM   | 0.144     | -5.37 | -5.65 | -8.16 | 2.37            | -5.79 | -1.41   | 0.058 | -3.07     | -4.00 | -6.57 | 2.56            | -4.01 | -0.97 | -0.086 | 2.30 | 1.65  | 1.59 | 0.19      | 1.78 | 0.44 |  |
| MRI-ESM2-0        | 0.155     | -7.53 | -8.04 | -9.86 | 1.73            | -8.13 | -1.36   | 0.068 | -3.82     | -4.29 | -4.14 | -0.50           | -4.64 | -0.54 | -0.087 | 3.71 | 3.75  | 5.72 | -2.23     | 3.49 | 0.82 |  |
| NorESM2-LM        | 0.051     | -3.06 | -4.23 | -5.48 | 2.06            | -3.42 | -1.21   | 0.032 | -1.27     | -2.56 | -3.67 | 1.01            | -2.66 | -0.78 | -0.020 | 1.79 | 1.66  | 1.81 | -1.06     | 0.76 | 0.42 |  |
| UKESM1-0-LL       | 0.126     | -4.40 | -5.08 | -8.31 | 3.20            | -5.11 | -1.98   | 0.069 | -0.82     | -3.19 | -6.48 | 3.07            | -3.41 | -0.88 | -0.057 | 3.58 | 1.89  | 1.83 | -0.14     | 1.70 | 1.10 |  |
| Ensemble (Mean)   | 0.112     | -4.43 | -5.39 | -7.76 | 2.28            | -5.49 | -1.49   | 0.062 | -2.06     | -3.51 | -5.52 | 1.70            | -3.82 | -0.82 | -0.050 | 2.37 | 1.89  | 2.24 | -0.57     | 1.67 | 0.67 |  |
| Ensemble (SD)     | 0.039     | 1.55  | 1.25  | 1.58  | 0.58            | 1.46  | 0.52    | 0.025 | 1.00      | 0.82  | 1.44  | 1.15            | 0.88  | 0.34  | 0.024  | 1.06 | 0.85  | 1.34 | 0.78      | 0.77 | 0.37 |  |

**Table 5.** As in Table 4, but for the winter (DJF).

|                   | 1970-1979 |       |       |       |                     |       |       |       | 2005-2014 |       |                     |      |       |       |        | Diff. |       |       |       |       |      |  |
|-------------------|-----------|-------|-------|-------|---------------------|-------|-------|-------|-----------|-------|---------------------|------|-------|-------|--------|-------|-------|-------|-------|-------|------|--|
| Model             | 4.00      |       | RF    | A     | ARC <sub>SURF</sub> |       | TAC   | AOD   | Е         | RF    | ARC <sub>SURF</sub> |      | TAC   | AOD   | ERF    |       | Α     | RCsu  |       | TAS   |      |  |
|                   | AOD       | TOA   | SURF  | SW    | LW                  |       | IAS   | AOD   | TOA       | SURF  | SW                  | LW   | NET   | IAS   | AOD    | TOA   | SURF  | SW    | LW    |       | IAS  |  |
| BCC-ESM1          | 0.119     | -     | -     | -3.48 | 1.08                | -2.40 | -1.21 | 0.110 | -         | -     | -3.51               | 1.92 | -1.59 | -0.98 | -0.009 | -     | -     | -0.03 | 0.84  | 0.81  | 0.23 |  |
| CNRM-ESM2-1       | 0.045     | -1.85 | -2.11 | -2.83 | 1.16                | -1.67 | -0.54 | 0.027 | -1.04     | -0.91 | -1.77               | 0.20 | -1.57 | 0.00  | -0.018 | 0.81  | 1.20  | 1.06  | -0.96 | 0.11  | 0.54 |  |
| EC-Earth3-AerChem | 0.058     | -4.60 | -2.53 | -8.26 | 2.87                | -5.39 | -2.75 | 0.024 | -1.33     | -1.88 | -3.96               | 1.78 | -2.17 | -0.76 | -0.034 | 3.28  | 0.65  | 4.30  | -1.09 | 3.21  | 1.99 |  |
| GFDL-ESM4         | 0.079     | -1.41 | -1.53 | -5.04 | 3.43                | -1.62 | -0.51 | 0.046 | -1.70     | -2.72 | -2.53               | 1.50 | -1.03 | -0.31 | -0.033 | -0.28 | -1.19 | 2.51  | -1.92 | 0.59  | 0.20 |  |
| MPI-ESM-1-2-HAM   | 0.205     | -6.06 | -4.03 | -8.17 | 4.04                | -4.13 | -1.01 | 0.090 | -5.33     | -3.21 | -3.05               | 0.55 | -2.50 | -0.57 | -0.115 | 0.73  | 0.82  | 5.12  | -3.49 | 1.63  | 0.44 |  |
| MRI-ESM2-0        | 0.122     | -7.17 | -4.80 | -6.72 | 3.40                | -3.32 | -0.82 | 0.054 | -2.32     | -2.26 | -3.04               | 1.39 | -1.65 | -0.03 | -0.068 | 4.85  | 2.54  | 3.68  | -2.02 | 1.67  | 0.79 |  |
| NorESM2-LM        | 0.035     | -1.94 | -1.85 | -3.68 | 2.25                | -1.43 | -0.69 | 0.042 | -0.77     | -0.63 | -4.23               | 1.77 | -2.46 | -0.54 | 0.007  | 1.16  | 1.22  | -0.55 | -0.47 | -1.03 | 0.15 |  |
| UKESM1-0-LL       | 0.090     | -3.44 | -2.84 | -5.12 | 3.32                | -1.81 | -1.61 | 0.046 | -0.50     | -1.58 | -5.10               | 3.90 | -1.20 | -0.60 | -0.043 | 2.94  | 1.26  | 0.02  | 0.58  | 0.61  | 1.01 |  |
| Ensemble (Mean)   | 0.094     | -3.78 | -2.81 | -5.41 | 2.69                | -2.72 | -1.14 | 0.055 | -1.86     | -1.89 | -3.40               | 1.63 | -1.77 | -0.47 | -0.039 | 1.93  | 0.93  | 2.01  | -1.07 | 0.95  | 0.67 |  |
| Ensemble (SD)     | 0.052     | 2.08  | 1.11  | 1.97  | 1.03                | 1.34  | 0.70  | 0.028 | 1.52      | 0.86  | 0.97                | 1.03 | 0.52  | 0.32  | 0.036  | 1.67  | 1.03  | 2.05  | 1.33  | 1.17  | 0.57 |  |

**Table 6.** As in Table 4, but for the summer (JJA).

|                   |       |       | 197    | 0-19                  | 79   |        |       | 2005-2014 |       |                      |        |                   |       |       |        | Diff. |      |              |       |      |       |  |
|-------------------|-------|-------|--------|-----------------------|------|--------|-------|-----------|-------|----------------------|--------|-------------------|-------|-------|--------|-------|------|--------------|-------|------|-------|--|
| Model             | AOD   |       | RF     | F ARC <sub>SURF</sub> |      |        | TAC   | AOD       |       | ERF ARC <sub>s</sub> |        | .RC <sub>su</sub> |       |       | AOD    | ERF   |      | $ARC_{SURF}$ |       |      | — TAS |  |
|                   | AOD   |       | SURF   | SW                    | LW   | NET    | TAS   | AOD       |       | SURF                 | SW     | LW                | NET   | IAS   | AOD    | TOA   | SURF | SW           | LW    | NET  | IAS   |  |
| BCC-ESM1          | 0.175 | -     | -      | -12.18                | 2.65 | -9.53  | -2.29 | 0.108     | -     | -                    | -6.42  | 1.14              | -5.28 | -2.01 | -0.067 | -     | -    | 5.76         | -1.52 | 4.25 | 0.27  |  |
| CNRM-ESM2-1       | 0.077 | -3.82 | -6.34  | -8.79                 | 2.87 | -5.92  | -1.00 | 0.032     | -1.27 | -2.93                | -5.32  | 2.11              | -3.21 | -0.30 | -0.044 | 2.55  | 3.41 | 3.47         | -0.76 | 2.71 | 0.70  |  |
| EC-Earth3-AerChem | 0.136 | -4.27 | -8.94  | -10.72                | 2.66 | -8.06  | -2.73 | 0.077     | 0.01  | -6.47                | -11.57 | 5.25              | -6.32 | -1.73 | -0.059 | 4.28  | 2.47 | -0.85        | 2.59  | 1.74 | 1.01  |  |
| GFDL-ESM4         | 0.127 | -2.79 | -8.01  | -11.66                | 3.30 | -8.36  | -1.63 | 0.072     | -1.34 | -4.87                | -7.99  | 1.39              | -6.60 | -1.01 | -0.056 | 1.45  | 3.13 | 3.67         | -1.91 | 1.75 | 0.62  |  |
| MPI-ESM-1-2-HAM   | 0.105 | -3.14 | -6.81  | -8.75                 | 2.29 | -6.45  | -1.94 | 0.034     | 0.84  | -3.82                | -6.59  | 2.50              | -4.08 | -1.25 | -0.071 | 3.98  | 2.99 | 2.16         | 0.21  | 2.37 | 0.68  |  |
| MRI-ESM2-0        | 0.195 | -8.15 | -11.07 | -12.36                | 0.52 | -11.84 | -1.53 | 0.068     | -4.77 | -7.08                | -5.33  | -1.48             | -6.81 | -0.97 | -0.127 | 3.37  | 3.99 | 7.03         | -2.00 | 5.03 | 0.56  |  |
| NorESM2-LM        | 0.074 | -2.65 | -5.56  | -6.15                 | 2.45 | -3.70  | -1.65 | 0.025     | -0.47 | -3.28                | -3.64  | 0.93              | -2.71 | -1.13 | -0.048 | 2.17  | 2.27 | 2.51         | -1.53 | 0.99 | 0.53  |  |
| UKESM1-0-LL       | 0.152 | -4.31 | -6.96  | -10.09                | 3.18 | -6.91  | -2.38 | 0.071     | 0.09  | -4.71                | -6.82  | 2.78              | -4.04 | -1.31 | -0.081 | 4.40  | 2.25 | 3.26         | -0.40 | 2.87 | 1.08  |  |
| Ensemble (Mean)   | 0.130 | -4.16 | -7.67  | -10.09                | 2.49 | -7.60  | -1.89 | 0.061     | -0.99 | -4.74                | -6.71  | 1.83              | -4.88 | -1.21 | -0.069 | 3.17  | 2.93 | 3.38         | -0.66 | 2.71 | 0.68  |  |
| Ensemble (SD)     | 0.041 | 1.74  | 1.72   | 1.98                  | 0.81 | 2.30   | 0.52  | 0.026     | 1.70  | 1.45                 | 2.20   | 1.79              | 1.49  | 0.48  | 0.025  | 1.05  | 0.60 | 2.22         | 1.43  | 1.26 | 0.24  |  |

**Table 7.** Annual and seasonal means for near-surface air temperature (TAS; in °C) over the Mediterranean region for the periods 1970-1979, 2005-2014 and their difference (2005-2014 minus 1970-1979). Values are presented for the three observational datasets and the multi-model ensembles for the CMIP6 "historical" and "hist-piAer" experiments. Only gridpoints over land were considered here.

| Dataset          |           | Annual    |       |           | DJF       |       | JJA       |           |       |  |  |  |
|------------------|-----------|-----------|-------|-----------|-----------|-------|-----------|-----------|-------|--|--|--|
| Dataset          | 1970-1979 | 2005-2014 | Diff. | 1970-1979 | 2005-2014 | Diff. | 1970-1979 | 2005-2014 | Diff. |  |  |  |
| Berkeley Earth   | 14.90     | 16.08     | 1.18  | 8.40      | 8.74      | 0.34  | 22.00     | 23.71     | 1.70  |  |  |  |
| CRU TS 4.08      | 14.78     | 15.92     | 1.14  | 6.99      | 7.27      | 0.27  | 23.06     | 24.75     | 1.68  |  |  |  |
| Uni. of Delaware | 14.33     | 15.53     | 1.20  | 5.90      | 6.11      | 0.21  | 23.33     | 25.13     | 1.81  |  |  |  |
| historical       | 14.45     | 15.79     | 1.34  | 6.29      | 7.48      | 1.19  | 23.15     | 24.82     | 1.67  |  |  |  |
| hist-piAer       | 16.04     | 16.72     | 0.68  | 7.48      | 7.97      | 0.48  | 25.23     | 26.25     | 1.02  |  |  |  |

630

635

**Figure 1.** Timeseries of the AA-induced evolution of aerosol optical depth (AOD; left column), effective radiative forcing (ERF; middle column), and atmospheric radiative cooling at the surface (ARC<sub>SURF</sub>; right column) over the Mediterranean region (10°W-40°E, 30°N-45°N) for the historical period (1850-2014) on an annual scale (top row) and in boreal winter (DJF; middle row) and summer (JJA; bottom row). Changes in AOD and ARC<sub>SURF</sub> were calculated from the difference "historical" minus "hist-piAer", whereas ERF from the difference "histSST" minus "histSST-piAer". The thin lines represent the annual weighted mean values, while thick lines represent the 10-year moving average. Shading expresses the inter-model variability (one standard deviation).

Figure 2. Changes in AOD due to anthropogenic aerosols over the Mediterranean (calculated from the difference "historical" minus "hist-piAer"). Spatial patterns are presented for the periods 1970-1979 (left column), 2005-2014 (middle column) and their difference (2005-2014 minus 1970-1979; right column) on an annual scale (top row) and in boreal winter (DJF; middle row) and summer (JJA; bottom row). The multi-model ensemble means over the Mediterranean region (shown as a box) are shown along with the inter-model variability (one standard deviation) at the top of each panel. Colored areas without markings indicate robust changes, while hatched (/) and cross-hatched (X) areas indicate non-robust changes and conflicting signals, respectively.

Figure 3. ERF<sub>TOA</sub> due to anthropogenic aerosols over the Mediterranean (calculated from the difference "histSST" minus "histSST-piAer"). Spatial patterns are presented for the periods 1970-1979 (left column), 2005-2014 (middle column) and their difference (2005-2014 minus 1970-1979; right column) on an annual scale (top row) and in boreal winter (DJF; middle row) and summer (JJA; bottom row). The multimodel ensemble means over the Mediterranean region (shown as a box) are shown along with the inter-model variability (one standard deviation) at the top of each panel. Colored areas without markings indicate robust changes, while hatched (/) and cross-hatched (X) areas indicate non-robust changes and conflicting signals, respectively.

**Figure 4.** Net ARC<sub>SURF</sub> due to anthropogenic aerosols over the Mediterranean (calculated from the difference "historical" minus "hist-piAer"). Spatial patterns are presented for the periods 1970-1979 (left column), 2005-2014 (middle column) and their difference (2005-2014 minus 1970-1979; right column) on an annual scale (top row) and in boreal winter (DJF; middle row) and summer (JJA; bottom row). The multi-model ensemble means over the Mediterranean region (shown as a box) are shown along with the inter-model variability (one standard deviation) at the top of each panel. Colored areas without markings indicate robust changes, while hatched (/) and cross-hatched (X) areas indicate non-robust changes and conflicting signals, respectively.

Figure 5. Timeseries of the AA-induced near-surface air temperature (TAS) evolution over the Mediterranean region (10°W-40°E, 30°N-45°N) for the historical period (1850-2014) on an annual scale (top row) and in boreal winter (DJF; middle row) and summer (JJA; bottom row). TAS changes were calculated from the difference "historical" minus "hist-piAer". The thin lines represent the annual weighted mean values, while thick lines represent the 10-year moving average. Shading indicates the inter-model variability expressed as one standard deviation.

**Figure 6.** Timeseries of land TAS evolution over the Mediterranean region (10°W-40°E, 30°N-45°N) for the historical period (1850-2014) as captured by CMIP6 ESMs and observations on an annual scale (top row) and in boreal winter (DJF; middle row) and summer (JJA; bottom row). Thick lines represent 10-year moving averages. In the cases of the CMIP6 experiments, the thin lines represent the annual weighted mean values, and shading indicates the inter-model variability expressed as one standard deviation.

Figure 7. Changes in TAS due to anthropogenic aerosols over the Mediterranean (calculated from the difference "historical" minus "hist-piAer"). Spatial patterns are presented for the periods 1970-1979 (left column), 2005-2014 (middle column) and their difference (2005-2014 minus 1970-1979; right column) on an annual scale (top row) and in boreal winter (DJF; middle row) and summer (JJA; bottom row). The multi-model ensemble means over the Mediterranean region (shown as a box) are shown along with the inter-model variability (one standard deviation) at the top of each panel. Colored areas without markings indicate robust changes, while hatched (/) and cross-hatched (X) areas indicate non-robust changes and conflicting signals, respectively.

**Figure 8.** Changes in zonal mean temperature due to AAs (calculated from the difference "historical" minus "hist-piAer") for the MED longitudes (10°W – 40°E). Cross sections (pressure level over latitude) are presented for the periods 1970-1979 (left column), 2005-2014 (middle column) and their difference (2005-2014 minus 1970-1979; right column) on an annual scale (top row) and in boreal winter (DJF; middle row) and summer (JJA; bottom row). Dashed vertical lines indicate the latitudinal boundaries of the MED region.

Figure 9. Changes in meridional mean temperature due to AAs (calculated from the difference "historical" minus "hist-piAer") for the MED latitudes (30°N – 45°N). Cross sections (pressure level over longitude) are presented for the periods 1970-1979 (left column), 2005-2014 (middle column) and their difference (2005-2014 minus 1970-1979; right column) on an annual scale (top row) and in boreal winter (DJF; middle row) and summer (JJA; bottom row). Dashed vertical lines indicate the longitudinal boundaries of the MED region.

Figure 10. Changes in geopotential height and winds at 850 hPa due to anthropogenic aerosols over the Mediterranean (calculated from the difference "historical" minus "hist-piAer"). Spatial patterns are presented for the periods 1970-1979 (left column), 2005-2014 (middle column) and their difference (2005-2014 minus 1970-1979; right column) on an annual scale (top row) and in boreal winter (DJF; middle row) and summer (JJA; bottom row). The multi-model ensemble means over the Mediterranean region (shown as a box) are shown along with the inter-model variability (one standard deviation) at the top of each panel. Colored areas without markings indicate robust changes, while hatched (/) and cross-hatched (X) areas indicate non-robust changes and conflicting signals, respectively.

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
