# Peer review of "Temperature and radiative responses to anthropogenic aerosols over the Mediterranean Basin based on CMIP6 Earth system models"

_EGUsphere, 2025_

## Referee Comment (RC2)

*"Temperature and radiative responses to anthropogenic aerosols over the Mediterranean based on CMIP6 Earth System models"*

**Authors: Alkiviadis Kalisoras, Prodromos Zanis, Aristeidis K. Georgoulias, Dimitris Akritidis, Robert J. Allen, Vaishali Naik**

This manuscript focuses on changes in temperature in response to anthropogenic aerosols (AA) reductions over the Mediterranean region. It includes two metrics of the radiative perturbation by AA, ARC and ERF, to examine the fast and slow temperature response respectively. The analysis is sound and of high quality. The main finding of the manuscript is that the increase in TAS due to AA reductions cannot be solely explained by reductions in AOD, but feedback between aerosol concentrations and circulation have also played a role. However, the manuscript in its current form is difficult to read due to very lengthy paragraphs, and lacking a red thread, without a clear motivation. Still these are issues that can be addressed and a revised version of this manuscript would be a good fit for a future publication in ACP.

**Essential comments**

**Introduction**

The introduction gives a good overview of the literature, however, it does not properly introduce the knowledge gaps that are addressed by the study. Accordingly, the motivation for the present study is not entirely apparent.

**Line 49 -73:** This paragraph lists the finding of several specific studies, however, it is not clear how each of these studies points towards the knowledge gap that the current study addresses, nor are these studies brought up again in the discussion or conclusion. Accordingly this paragraph could likely be shortened and focused toward the knowledge gaps that will be addressed through the objectives defined in the next paragraph. For example; what do the previous studies miss by not examining the fast and slow temperature responses to AA reductions jointly?

**Data and Methodology**

The data and methodology covers the metrics and datasets used. However, the text could benefit from being more concise and understandable. For instance, rather than citing Eyring et al., 2016 and Collins et al 2017, repeatedly when referring to AerChemMIP and CMIP6, just use the name of the MIP instead of the reference.

When referring to names of the different model experiments, it would be more appropriate to use italic to emphasize the names rather than double quotes.

**On line 104-105:** It is not necessary to repeat that AerChemMIP has fixed SSTs and sea ice concentration. Just mention this one when the AerChemMIP simulations are introduced on line 100.

**On line 126-138:** Avoid repeating the definition of ERF several times within the same paragraph.

**On line 149:** ARC is defined in the manuscript as "*ARC is defined as the net shortwave (SW) and longwave (LW) radiation loss of the atmospheric column*", which is the common definition in the literature. Accordingly, from this definition, ARC should be the difference between the net radiative flux at the top-of-the-atmosphere and surface. However in Eq. 3 the authors introduce a new terminology of $ARC_{TOA}^{NET}$ and $ARC_{SURF}^{NET}$, which in the manuscript simply refers to the net radiative flux at TOA and surface respectively when calculated from the hist-pier/ historically fully coupled simulations. Using the term ARC in this new context is confusing, since this is not actually radiative loss of the atmosphere. Furthermore, ARC does not strictly imply that it is derived from a coupled fully coupled simulation, as it is common to compare ARC from fSST simulations to examine fast versus slow precipitation responses. Therefore, it would be advisable to use a different terminology than $ARC_{TOA}^{NET}$ and $ARC_{SURF}^{NET}$ referring to radiative perturbations in this manuscript.

**Results**

Overall the figures and results are well described and of good quality. However, due to very information dense and lengthy paragraphs that often span several figures and sometimes going back and forth between them, the section overall is very challenging to read. This was not helped by several paragraphs ending without any particular conclusion. The authors are advised to closely re-examine each paragraph and provide a red thread for the reader to follow.

**3.1 Radiative changes**

The title of the subsection is not very informative, so it would be advisable to choose a title that relates to a research question.

**Line 180 - 183:**  *AA concentrations peak__ed__ in the late 1970s to early 1980s in Europe and the MED. Consequently, AOD change reaches a maximum during that period (Fig. 1, left column). The AOD maximum*  *is larger in magnitude during the boreal summer (JJA) than the winter (DJF). Likewise, __the__ transient __annual mean__ ERF both at TOA and surface attained its most negative values during the late 1970s - early 1980s* *, dominated by the evolution of ERF in JJA (Fig. 1, middle column).*

**Line 183-185:** Not sure what is the new information in this sentence, perhaps remove?

**Line 185-189:** Here, one example of text that is very difficult for a reader to understand due to jumping between topics within the same sentence, e.g. inter-model spread, temporal evolution, magnitude of the ERF peak and ARC. This makes the paragraph as a whole difficult to comprehend. This could be simplified by for instance only describing the ERF and then in the next paragraph describing ARC.

**Line 207-209:** The first sentence sets the expectation that the spatial pattern of ERF will be described next, then the reader would expect that the following sentence would build on this. However here the reader gets led astray when this sentence is followed by a listing of more tables and figures.

**Line 233 -268:** This paragraph needs to be broken down into smaller paragraphs.

**3.2 Temperature changes**

Again, the title of this section is not very informative. The section is started off by another massive paragraph, which is difficult to navigate. Overall, this section is as challenging to read as the previous section.

It would also benefit more from a discussion that compares the findings of this study with previous studies.

**4. Conclusion**

The conclusion gives a good summary of the findings of the study, however, what is lacking is the context from the literature.

The conclusion does not fully answer the objective of the manuscript defined in the introduction, in particular with respect to quantifying the fast and total radiative response of global changes in AA emission. If this is written in the introduction, then there is an expectation that this will be brought up again in the conclusions.

**Minor typographical comment:**

The manuscript uses hyphens in places where an en dash is typographically correct (for example "1980-2000"). Please replace hyphens with en dashes for numeric ranges (1980–2000), similarly hyphens should not be used for minus signs. Use hyphens only for compound words and em dashes for clause breaks.

---

## Author Comment (AC2)

**Reply to O. Haugvaldstad (Reviewer #2)**

We would like to thank O. Haugvaldstad for the constructive and helpful comments. The reviewer's contribution is recognized in the acknowledgments of the revised manuscript. Below follows our response point by point. The reviewer's comments are given in *italic* and our response is given in **bold** font.

Essential comments:

Introduction

*1)* The Reviewer notes: "*The introduction gives a good overview of the literature, however, it does not properly introduce the knowledge gaps that are addressed by the study. Accordingly, the motivation for the present study is not entirely apparent.*"

**Our study uses an ensemble of state-of-the-art CMIP6 Earth system models to provide a basin-wide quantification of the Mediterranean warming attributable to global anthropogenic aerosol emission reductions. Previous studies often focused on individual subregions of the Euro-Mediterranean region or used data derived from other sources (reanalysis products or regional climate models). Moreover, this work fills a methodological gap by connecting ERF (fast forcing) to ARC (total atmospheric energy adjustment) and then to vertical and horizontal total temperature responses and circulation responses. This study also addresses the performance of CMIP6 models in simulating the observed annual and seasonal land temperature trends over the Mediterranean by comparing them to observational datasets. This has been one of our main motivations as there is only little literature investigating this aspect. The Introduction was updated to address the reviewer's concerns.**

*2)* The Reviewer notes: "*Line 49 -73: This paragraph lists the finding of several specific studies, however, it is not clear how each of these studies points towards the knowledge gap that the current study addresses, nor are these studies brought up again in the discussion or conclusion. Accordingly this paragraph could likely be shortened and focused toward the knowledge gaps that will be addressed through the objectives defined in the next paragraph. For example; what do the previous studies miss by not examining the fast and slow temperature responses to AA reductions jointly?*"

**A new paragraph has been added in the Introduction focusing on the knowledge gaps from previous studies that are investigated in our study (see also our reply to Comment #1):**

**"Despite the plethora of studies investigating aerosol-driven changes over the Euro-Mediterranean region, there still remain several knowledge gaps: a) how do global climate models simulate the Mediterranean response to global anthropogenic aerosol emission changes on basin-wide scale, b) how do changes in aerosol optical depth, the radiative budget, and atmospheric circulation translate to horizontal and vertical temperature changes over the Mediterranean, c) how well do climate models perform when simulating the observed land temperature amplification on annual and seasonal scales?"**

Data and Methodology

*3)* The Reviewer notes: "*The data and methodology covers the metrics and datasets used. However, the text could benefit from being more concise and understandable. For instance, rather than citing Eyring et al., 2016 and Collins et al 2017, repeatedly when referring to AerChemMIP and CMIP6, just use the name of the MIP instead of the reference.*"

**The text was revised accordingly as suggested by the reviewer.**

*4)* The Reviewer notes: "*When referring to names of the different model experiments, it would be more appropriate to use italic to emphasize the names rather than double quotes.*"

**The manuscript (including table and figure captions) was revised accordingly as suggested by the reviewer.**

*5)* The Reviewer notes: "*On line 104-105: It is not necessary to repeat that AerChemMIP has fixed SSTs and sea ice concentration. Just mention this one when the AerChemMIP simulations are introduced on line 100.*"

**Lines 104-105 expand on the definition of prescribed SSTs and are not a simple repetition. It is essential that the *historical* experiment is described before it is explained that "SSTs and sea ice are prescribed to the monthly mean time-evolving values from the corresponding *historical* simulation of each model".**

*6)* The Reviewer notes: "*On line 126-138: Avoid repeating the definition of ERF several times within the same paragraph.*"

**It was revised accordingly as suggested by the reviewer.**

*7)* The Reviewer notes: "*On line 149: ARC is defined in the manuscript as "ARC is defined as the net shortwave (SW) and longwave (LW) radiation loss of the atmospheric column", which is the common definition in the literature. Accordingly, from this definition, ARC should be the difference between the net radiative flux at the top-of-the-atmosphere and surface. However in Eq. 3 the authors introduce a new terminology of $ARC_{TOA}^{NET}$ and $ARC_{SURF}^{NET}$, which in the manuscript simply refers to the net radiative flux at TOA and surface respectively when calculated from the hist-pier/ historically fully coupled simulations. Using the term ARC in this new context is confusing, since this is not actually radiative loss of the atmosphere. Furthermore, ARC does not strictly imply that it is derived from a coupled fully coupled simulation, as it is common to compare ARC from fSST simulations to examine fast versus slow precipitation responses. Therefore, it would be advisable to use a different terminology than $ARC_{TOA}^{NET}$ and $ARC_{SURF}^{NET}$ referring to radiative perturbations in this manuscript.*"

**It is true that ARC does not need to be calculated strictly from coupled simulations. However, in our study we used ERF as a standard metric for the fast forcing due to anthropogenic aerosols, which (by definition) is calculated only from prescribed/fixed-SST simulations. To investigate the total forcing due to anthropogenic aerosols we therefore used another standard metric — ARC — calculated from coupled simulations because we already used ERF to calculate fast responses to the radiative budget. We did**

not intend to imply that ARC can only be calculated from coupled simulations. This paragraph was updated to clarify the above.

**When it comes to the definition of ARC, we used it as a standard metric of the radiative loss of the atmosphere without deviating from its common definition in existing literature. We do not introduce a new terminology of $ARC_{TOA}^{NET}$ and $ARC_{SURF}^{NET}$; we merely consider the energy flux as positive when exiting the atmospheric column (i.e., upward positive at TOA and downward positive at the surface). In previous studies SW and LW fluxes were considered positive towards opposite directions both at TOA and surface, and hence the need for subtraction (i.e., "the difference between the net radiative flux at the top-of-the-atmosphere and surface"). In Eq. 3, we calculated the sum of $ARC_{TOA}^{NET}$ and $ARC_{SURF}^{NET}$ as the former is positive upwards and the latter positive downwards. We simply changed the sign of one term without changing its physical interpretation. We believe that it would be easier for the reader to follow the discussion on ARC changes if both quantities were consistently considered positive when exiting the atmospheric column. This was clarified in the main text.**

Results

*8)* The Reviewer notes: "*Overall the figures and results are well described and of good quality. However, due to very information dense and lengthy paragraphs that often span several figures and sometimes going back and forth between them, the section overall is very challenging to read. This was not helped by several paragraphs ending without any particular conclusion. The authors are advised to closely re-examine each paragraph and provide a red thread for the reader to follow.*"

**The Results section was modified to provide a more linear reading of the manuscript.**

3.1 Radiative changes

*9)* The Reviewer notes: "*The title of the subsection is not very informative, so it would be advisable to choose a title that relates to a research question.*"

**We divided the Results section into more subsections where discrete topics are discussed.**

*10)* The Reviewer notes: "*Line 180 - 183:  AA concentrations peaked in the late 1970s to early 1980s in Europe and the MED. Consequently, AOD change reaches a maximum during that period (Fig. 1, left column). The AOD maximum  is larger in magnitude during the boreal summer (JJA) than the winter (DJF). Likewise, the transient annual mean ERF both at TOA and surface attained its most negative values during the late 1970s - early 1980s , dominated by the evolution of ERF in JJA (Fig. 1, middle column).*"

**It was revised accordingly as suggested by the reviewer.**

*11)* The Reviewer notes: "*Line 183-185: Not sure what is the new information in this sentence, perhaps remove?*"

**The new information is that ERF$_{SURF}$ is weaker than ERF$_{TOA}$ during the winter of 1975 – 1985 (in contrast to JJA and annual ERF$_{SURF}$). We have clarified this in the main text.**

*12)* The Reviewer notes: "*Line 185-189: Here, one example of text that is very difficult for a reader to understand due to jumping between topics within the same sentence, e.g. inter-model spread, temporal evolution, magnitude of the ERF peak and ARC. This makes the paragraph as a whole difficult to comprehend. This could be simplified by for instance only describing the ERF and then in the next paragraph describing ARC.*"

**It was revised accordingly as suggested by the reviewer.**

*13)* The Reviewer notes: "*Line 207-209: The first sentence sets the expectation that the spatial pattern of ERF will be described next, then the reader would expect that the following sentence would build on this. However here the reader gets led astray when this sentence is followed by a listing of more tables and figures.*"

**The second sentence ("The annual area-weighted … and 6 (for JJA).") was removed.**

*14)* The Reviewer notes: "*Line 233 -268: This paragraph needs to be broken down into smaller paragraphs.*"

**It was revised accordingly as suggested by the reviewer.**

3.2 Temperature changes

*15)* The Reviewer notes: "*Again, the title of this section is not very informative. The section is started off by another massive paragraph, which is difficult to navigate. Overall, this section is as challenging to read as the previous section.*"

**Same as Comment #9.**

*16)* The Reviewer notes: "*It would also benefit more from a discussion that compares the findings of this study with previous studies.*"

**It was revised accordingly as suggested by the reviewer.**

4. Conclusion

*17)* The Reviewer notes: "*The conclusion gives a good summary of the findings of the study, however, what is lacking is the context from the literature. The conclusion does not fully answer the objective of the manuscript defined in the introduction, in particular with respect to quantifying the fast and total radiative response of global changes in AA emission. If this is*

*written in the introduction, then there is an expectation that this will be brought up again in the conclusions.*"

**We agree with the reviewer and have revised the Conclusions section to explicitly reconnect with the objectives stated in the Introduction. We now clearly summarize and quantify both the fast radiative response (ERF) and the total atmospheric energy response (ARC) to global anthropogenic aerosol emission changes, and place these results in the context of previous studies focusing on aerosol-driven changes over the Mediterranean.**

Minor typographical comment:

*18)* The Reviewer notes: "*The manuscript uses hyphens in places where an en dash is typographically correct (for example "1980-2000"). Please replace hyphens with en dashes for numeric ranges (1980–2000), similarly hyphens should not be used for minus signs. Use hyphens only for compound words and em dashes for clause breaks.*"

**The manuscript (including tables and figures) was revised accordingly as suggested by the reviewer. En dashes are now used for numeric ranges and minus signs.**